# Parp1 hyperactivity couples DNA breaks to aberrant neuronal calcium signalling and lethal seizures

Emilia Komulainen[1], Jack Badman[1,2], Stephanie Rey[2], Stuart Rulten[1], Limei Ju[1], Kate Fennell[2], Ilona Kalasova[3] (iD), Kristyna Ilievova[3], Peter J McKinnon[4] (iD), Hana Hanzlikova[1,3] (iD), Kevin Staras[2,*] (iD) & Keith W Caldecott[1,3,**] (iD)

## Abstract

Defects in DNA single-strand break repair (SSBR) are linked with neurological dysfunction but the underlying mechanisms remain poorly understood. Here, we show that hyperactivity of the DNA strand break sensor protein Parp1 in mice in which the central SSBR protein Xrcc1 is conditionally deleted (*Xrcc1*[*Nes-Cre*]) results in lethal seizures and shortened lifespan. Using electrophysiological recording and synaptic imaging approaches, we demonstrate that aberrant Parp1 activation triggers seizure-like activity in Xrcc1-defective hippocampus *ex vivo* and deregulated presynaptic calcium signalling in isolated hippocampal neurons *in vitro*. Moreover, we show that these defects are prevented by Parp1 inhibition or deletion and, in the case of Parp1 deletion, that the lifespan of *Xrcc1*[*Nes-Cre*] mice is greatly extended. This is the first demonstration that lethal seizures can be triggered by aberrant Parp1 activity at unrepaired SSBs, highlighting PARP inhibition as a possible therapeutic approach in hereditary neurological disease.

**Keywords** DNA strand break; neurodegeneration; poly(ADP-ribose) polymerase; seizures; XRCC1

**Subject Categories** DNA Replication, Recombination & Repair; Molecular Biology of Disease; Neuroscience

## Introduction

DNA single-strand breaks (SSBs) are the commonest DNA lesions arising in cells and are rapidly detected by poly(ADP-ribose) polymerase-1 (PARP1) and/or poly(ADP-ribose) polymerase-2 (PARP2), enzymes that are activated at DNA breaks and modify themselves and other proteins with mono-ADP-ribose and/or poly-ADP-ribose (Benjamin & Gill, 1980; Chaudhuri & Nussenzweig, 2017; Hanzlikova *et al*, 2017; Azarm & Smith, 2020). Poly(ADP-ribose) triggers recruitment of the DNA single-strand break repair (SSBR) scaffold protein XRCC1 and its protein partners to facilitate the repair of SSBs (Breslin *et al*, 2015; Hanzlikova *et al*, 2017; Caldecott, 2019). If not repaired rapidly, SSBs can result in replication fork stalling and/or collapse and can block the progression of RNA polymerases during gene transcription (Hsiang *et al*, 1989; Ryan *et al*, 1991; Zhou & Doetsch, 1993; Kuzminov, 2001; Kathe *et al*, 2004). Notably, mutations in proteins involved in SSBR in humans are associated with cerebellar ataxia, neurodevelopmental defects and episodic seizures (Caldecott, 2008; Yoon & Caldecott, 2018). To date, all identified SSBR-defective human diseases are mutated in either XRCC1 or one of its protein partners (Caldecott, 2019).

Recently, we demonstrated using an Xrcc1-defective mouse model (*Xrcc1*[*Nes-Cre*]) that SSBR-defective cerebellum possesses elevated steady-state levels of poly(ADP-ribose) resulting from the hyperactivation of Parp1, leading to the loss of cerebellar interneurons and cerebellar ataxia (Lee *et al*, 2009; Hoch *et al*, 2017). PARP1 hyperactivity can trigger cellular dysfunction and/or cytotoxicity by several mechanisms including excessive depletion of NAD$^+$/ATP and/or by generating excessive amounts of poly(ADP-ribose) (Zhang *et al*, 1994; Andrabi *et al*, 2006; Yu *et al*, 2006; Andrabi *et al*, 2014). However, the extent to which Parp1 hyperactivation might account for the spectrum of neurological pathology induced by unrepaired endogenous SSBs is unknown. Here, we have addressed this question. We show that aberrant Parp1 activity extends beyond the cerebellum in *Xrcc1*[*Nes-Cre*] mice and is evident across the brain, resulting in deregulated neuronal presynaptic calcium signalling, lethal seizures and shortened lifespan. Importantly, we demonstrate that Parp1 inhibition or deletion prevents the Ca$^{2+}$ signalling defects and elevated seizure-like activity in *Xrcc1*[*Nes-Cre*] hippocampus and that Parp1 deletion prolongs the

---

1   Genome Damage and Stability Centre, School of Life Sciences, University of Sussex, Brighton, UK
2   Sussex Neuroscience, School of Life Sciences, University of Sussex, Brighton, UK
3   Department of Genome Dynamics, Institute of Molecular Genetics of the Czech Academy of Sciences, Prague, Czech Republic
4   Department of Genetics, St Jude Children's Research Hospital, Memphis, TN, USA
    *Corresponding author. Tel: +44 01273 678478; E-mail: k.staras@sussex.ac.uk
    **Corresponding author. Tel: +44 01273 877519; E-mail: k.w.caldecott@sussex.ac.uk

lifespan of $Xrcc1^{Nes-Cre}$ mice. These findings highlight the potential of Parp1 as a target in the therapeutic treatment of XRCC1-defective disease.

# Results

## Parp1 is hyperactive throughout Xrcc1$^{Nes-Cre}$ brain

We reported previously that Parp1 is hyperactive in the cerebellum of $Xrcc1^{Nes-Cre}$ mice, resulting in cerebellar ataxia (Hoch *et al*, 2017). Here, to extend this, we measured the steady-state level of pan-ADP-ribose signal across Xrcc1-defective brain by immunohistochemistry. We detected increased levels of ADP-ribose throughout $Xrcc1^{Nes-Cre}$ brain, including the cortex, with particularly strong immunostaining in the cerebellum and hippocampus (Fig 1A). In contrast, we did not detect elevated levels of Atm protein, an unrelated DNA repair-associated antigen, ruling out that the elevated anti-ADP-ribose signal was a non-specific artefact (Fig EV1). Indeed, we confirmed that the elevated ADP-ribose signal was the product of endogenous Parp1 activity, because *Parp1* deletion in $Xrcc1^{Nes-Cre}$ mice reduced this signal to levels below those in wild-type brain (Fig 1A; $Parp1^{-/-}/Xrcc1^{Nes-Cre}$). Consistent with this, the elevated ADP-ribose signal in $Xrcc1^{Nes-Cre}$ brain was also detected if we employed antibody specific for poly(ADP-ribose), which is the primary ADP-ribosylation product of Parp1 activity (Fig EV1).

To examine whether we could recapitulate the increased ADP-ribosylation in $Xrcc1^{Nes-Cre}$ brain biochemically, we incubated tissue extracts from wild-type and $Xrcc1^{Nes-Cre}$ forebrain, containing cortex and hippocampus, with $NAD^+$ to stimulate ADP-ribosylation *in vitro*. Indeed, wild-type but not $Parp1^{-/-}$ forebrain extracts rapidly accumulated ADP-ribosylated proteins when incubated with $NAD^+$ (Fig 1B). More importantly, tissue extracts prepared from $Xrcc1^{Nes-Cre}$ forebrain accumulated ADP-ribosylated proteins more rapidly and to a greater extent than did wild-type extracts, and similar results were observed if we employed tissue extracts from cerebellum (Fig 1B and C). Consistent with ongoing Parp1 hyperactivity, the steady-state level of $NAD^+$ in $Xrcc1^{Nes-Cre}$ forebrain was half that present in wild-type forebrain and was increased by deletion of even a single Parp1 allele (Fig 1D). Collectively, these data implicate widespread Parp1 hyperactivation in Xrcc1-defective brain, presumably as a result of the underlying defect in SSB repair (Lee *et al*, 2009). In agreement with this, we did not detect elevated ADP-ribose in brain from $Ku70^{-/-}$ mice in which the primary pathway for DNA double-strand break (DSB) repair in brain is defective, confirming that the elevated ADP-ribose in $Xrcc1^{Nes-cre}$ brain was the result of unrepaired SSBs, not DSBs (Fig 1A).

## Parp1 hyperactivation triggers juvenile seizures and mortality in Xrcc1$^{Nes-Cre}$ mice

Next, we generated Kaplan–Meier survival curves to determine the influence of *Xrcc1* on lifespan. As expected (Lee *et al*, 2009), $Xrcc1^{Nes-Cre}$ animals exhibited greatly reduced longevity when compared to wild-type mice, with the cohort employed in these experiments having a median lifespan of ~ 3–4 weeks (Fig 2A). Given the widespread Parp1 hyperactivity across the brain in $Xrcc1^{Nes-Cre}$ mice, we examined the impact of Parp1 deletion on

lifespan. Remarkably, the median lifespan of $Parp1^{+/-}/Xrcc1^{Nes-Cre}$ and $Parp^{-/-}/Xrcc1^{Nes-Cre}$ littermates in which one or both *Parp1* alleles were additionally deleted was increased ~ 25-fold and ~ 7-fold, to 79 and 23 weeks, respectively (Fig 2A). This result demonstrates that aberrant Parp1 activity in Xrcc1-defective brain is a major contributor to organismal death. It is noteworthy that deletion of one *Parp1* allele prolonged the lifespan of $Xrcc1^{Nes-Cre}$ mice to a greater extent than deletion of both *Parp1* alleles. This suggests that whilst the loss of a single *Parp1* allele is sufficient to suppress Parp1-induced toxicity, the loss of the second allele eradicates an additional role for Parp1 in Xrcc1-defective brain that is important for survival.

Next, to establish the cause of Parp1-dependent death in $Xrcc1^{Nes-Cre}$ mice, we conducted infrared video imaging for a four-day period starting at P15. These experiments revealed that $Xrcc1^{Nes-Cre}$ mice experienced sporadic seizures, culminating ultimately in a lethal seizure from which animals did not recover (Fig 2B). In contrast, we did not observe any seizures in wild-type mice over the same time period. The cause of death induced by the seizures is unclear but, similar to sudden unexpected death during epilepsy (SUDEP) in humans, it is likely to result from the disruption of normal cardiac or respiratory function (Surges *et al*, 2009). Importantly, we also did not detect seizures in $Xrcc1^{Nes-Cre}$ mice in which one or both alleles of Parp1 were deleted over the time course of the experiment, consistent with the increased lifespan of these mice (Fig 2B). To our knowledge, this is the first demonstration that seizures can be triggered by aberrant Parp1 activity.

## Seizure-like activity in Xrcc1$^{Nes-Cre}$ brain slices is corrected by Parp1 deletion

The induction of seizures by Parp1 in $Xrcc1^{Nes-Cre}$ mice is consistent with the strikingly high level of poly(ADP-ribose) in the hippocampus of these animals, because defects in this region of the brain are often associated with seizure activity (Gunn & Baram, 2017). To examine directly whether elevated seizure-like activity is present in $Xrcc1^{Nes-Cre}$ hippocampus, we carried out targeted extracellular electrophysiological recording experiments in brain slices prepared from wild-type mice, $Xrcc1^{Nes-Cre}$ mice and $Xrcc1^{Nes-Cre}$ mice in which *Parp1* was additionally deleted. When the brain slices were washed into an epileptogenic solution, the mean cumulative number of seizure-like events in the CA3 region of $Xrcc1^{Nes-Cre}$ hippocampus was ~ 2.5-fold greater than in wild-type hippocampus (Fig 3A–C). Moreover, strikingly, this elevated seizure-like activity was reduced or prevented if one or both alleles of *Parp1* were deleted, respectively (Fig 3A–C), confirming Parp1 as the cause of the elevated seizure activity in $Xrcc1^{-/-}$ hippocampus.

To extend these analyses, we employed a high-density multielectrode array (HD-MEA) platform (Fig 4A). This approach allowed us to assay the spatial organization of network activity across hippocampal and cortical structures simultaneously and with high temporal resolution. In particular, we recorded the onset and threshold of seizure-like events in the cortex, CA1 and CA3 regions of brain slices perfused into an epileptogenic solution, and plotted the cumulative activity in each. In agreement with the results described above, we found that seizure-like events were significantly higher in $Xrcc1^{Nes-Cre}$ cortex and hippocampus when compared to wild type, with the strongest effects observed in the hippocampal CA3 region (Fig 4B–D).

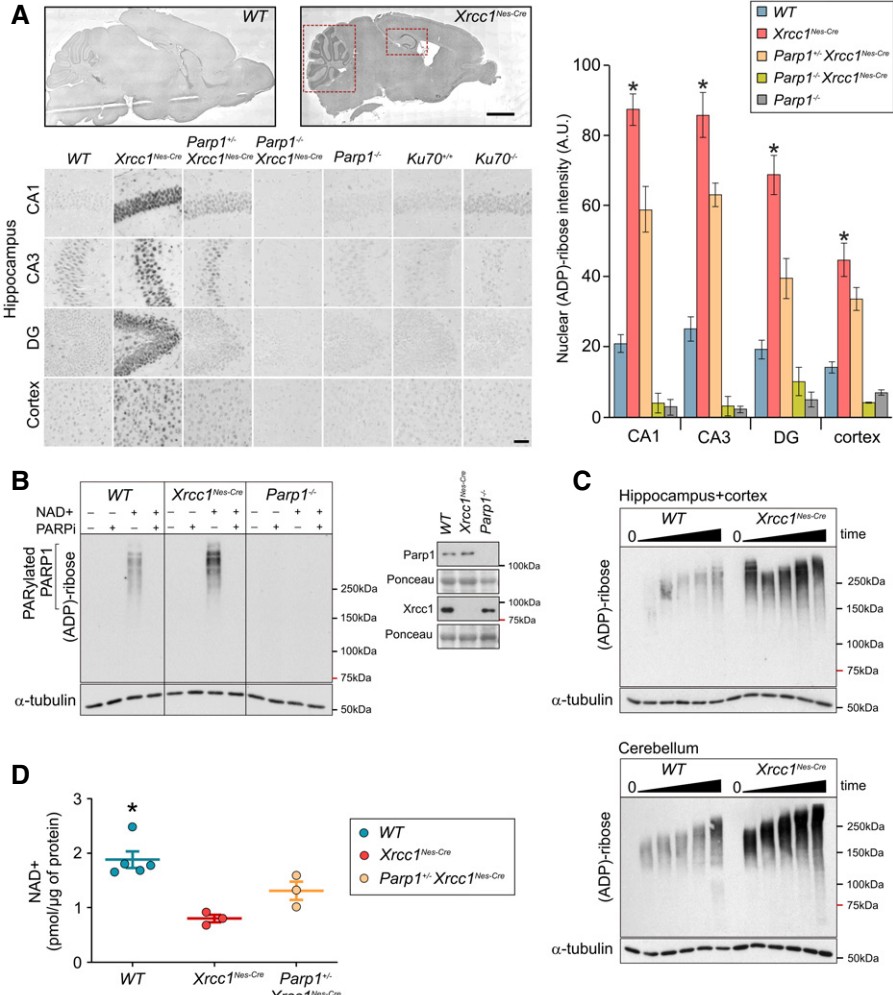

**Figure 1. Hyperactivity of Parp1 in Xrcc1$^{Nes-Cre}$ brain.**

A   Sagittal sections obtained from mice (p15) of the indicated genotypes were immunostained for ADP-ribose using the pan-ADP-ribose detection reagent MABE1016. Representative images showing levels of ADP-ribose in the hippocampal regions CA1, CA3 and dentate gyrus (DG), and in the cerebral cortex. Red dotted boxes highlight the elevated ADP-ribose staining in Xrcc1$^{Nes-Cre}$ cerebellum (left dotted box) and hippocampus (right doted box). Scale bars: 5 mm, 50 μm. WT (n = 4 mice), Xrcc1$^{Nes-Cre}$ (n = 4), Parp1$^{+/-}$ /Xrcc1$^{Nes-Cre}$ (n = 4), Parp1$^{-/-}$/Xrcc1$^{Nes-Cre}$ (n = 3) and Parp1$^{-/-}$ (n = 3). Summary histograms show mean ± SEM. Pairwise comparisons between WT versus Xrcc1$^{Nes-Cre}$ mice and WT versus Parp1$^{-/-}$/Xrcc1$^{Nes-Cre}$ mice were conducted by Kruskal–Wallis ANOVA with Dunn's post hoc test, and statistically significant differences (*P < 0.05) are shown.

B   Protein extracts from wild-type (WT), Xrcc1$^{Nes-Cre}$ and Parp1$^{-/-}$ forebrain tissue, containing hippocampus and cortex, were incubated with 1 mM NAD$^+$ for 45 min in the presence or absence of PARP inhibitor as indicated, and ADP-ribosylation detected by Western blotting using the poly(ADP-ribose)-specific detection reagent MABE1031. Representative images from two or more independent experiments are shown. A Western blot showing the level of Parp1 and Xrcc1 in the forebrain tissue extracts is also shown, Inset.

C   Protein extracts from wild-type (WT) and Xrcc1$^{Nes-Cre}$ forebrain or cerebellum were incubated for 0, 5, 10, 15, 30 and 45 min in the presence of NAD$^+$ as above.

D   NAD$^+$ levels in wild-type (WT), Xrcc1$^{Nes-Cre}$ and Parp$^{+/-}$/Xrcc1$^{Nes-Cre}$ forebrain tissue. Data are the scatterplots of individual measurements from at least three mice per genotype, with error bars representing the mean ± SEM. Statistically significant differences with Xrcc1$^{Nes-cre}$ are indicated (Kruskal–Wallis ANOVA with Dunn's post hoc test *P < 0.05).

Given the ability of Parp1 deletion to rescue normal levels of seizure-like activity in Xrcc1$^{Nes-Cre}$ brain, we extended these experiments to examine the impact of PARP1 inhibitor. To do this, we administered PARP1 inhibitor (ABT-888; veliparib) ad libitum in the drinking water from day 10 until their analysis at days 14–17. Strikingly, whereas this application of PARP1 inhibitor had little effect on the seizure-like activity of brain sections prepared from wild-type mice, it ablated the elevated seizure-like activity in Xrcc1$^{Nes-Cre}$ brain slices, in all three regions of the brain tested (Fig 4E). Together,

these data confirm that Parp1 hyperactivity in Xrcc1$^{Nes-Cre}$ brain triggers increased seizure-like activity that can be prevented by Parp1 deletion or pharmacological inhibition.

## Aberrant Parp1 activity deregulates presynaptic calcium signalling in Xrcc1$^{Nes-Cre}$ neurons

It is currently unclear how aberrant Parp1 activity at unrepaired SSBs might trigger seizures. However, it is known that Parp1

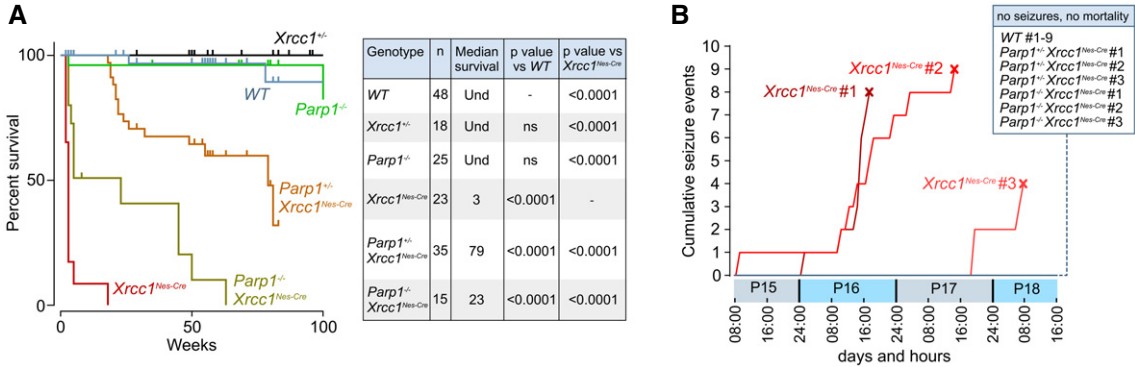

**Figure 2. Parp1 hyperactivation triggers juvenile seizures and mortality in the absence of Xrcc1.**

A   Kaplan–Meier curves for survival of mice of the indicated genotypes. The table shows number of individuals in each group, median survival (weeks) and *P*-values from pairwise curve comparisons; Und: undetermined (log-rank Mantel–Cox tests).

B   Video monitoring and recording of generalized running/bouncing seizures in mice of the indicated genotypes from P15-P19. The point of death of *Xrcc1*<sup>Nes-Cre</sup> mice by fatal seizure is indicated (cross). Note no seizures from mice of other genotypes were detected. *WT* (n = 9 mice), *Xrcc1*<sup>Nes-cre</sup> (n = 3 mice), *Parp1*<sup>+/−</sup>/*Xrcc1*<sup>Nes-Cre</sup> (n = 3 mice) and *Parp1*<sup>−/−</sup>/*Xrcc1*<sup>Nes-Cre</sup> (n = 3).

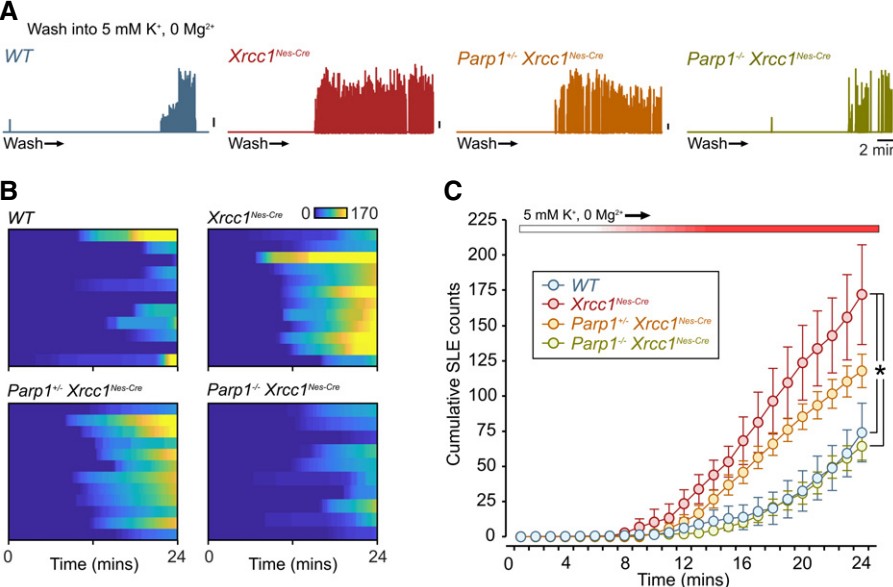

**Figure 3. Suppression of seizure-like activity in *Xrcc1*<sup>Nes-Cre</sup> brain slices by *Parp1* deletion.**

A   Plots of enveloped spike activity waveforms based on targeted extracellular recordings from hippocampal CA3 region perfused into epileptogenic buffer for the indicated genotypes. Upward deflections indicate seizure-like episodes (SLEs). Scale bars, 5% of maximum amplitude.

B   Heatplots of cumulative SLEs: *WT* (n = 11 slices from three mice), *Xrcc1*<sup>Nes-Cre</sup> (n = 12 slices from four mice), *Parp1*<sup>+/−</sup>/*Xrcc1*<sup>Nes-Cre</sup> (n = 12 slices from three mice) and *Parp1*<sup>−/−</sup>/*Xrcc1*<sup>Nes-Cre</sup> (n = 10 slices from three mice). Each horizontal bar corresponds to one slice with the colour code indicating cumulative SLEs.

C   Summary of mean ± SEM cumulative SLE counts. The number of replicates was the same as in (B). At the recording endpoint, the total SLE count in *Xrcc1*<sup>Nes-Cre</sup> is significantly higher than both *WT* and *Parp1*<sup>−/−</sup> *Xrcc1*<sup>Nes-Cre</sup> (Kruskal–Wallis ANOVA, $P < 0.0017$, * indicates significance with Dunn's *post hoc* comparisons).

activity can affect the expression of many genes that might influence seizure activity, including those affecting $Ca^{2+}$ homeostasis (Stoyas *et al*, 2019). Consequently, we examined whether the seizure-like activity in *Xrcc1*<sup>Nes-Cre</sup> mice reflects a defect in $Ca^{2+}$ signalling at the level of single synapses in isolated hippocampal neurons. Similar to

whole brain sections, we detected elevated endogenous levels of ADP-ribosylation in isolated *Xrcc1*<sup>Nes-Cre</sup> neurons, although as observed previously in other cultured cell types (Hanzlikova *et al*, 2018) the detection of endogenous poly(ADP-ribose) required incubation for 1 h with an inhibitor of poly(ADP-ribose) glycohydrolase

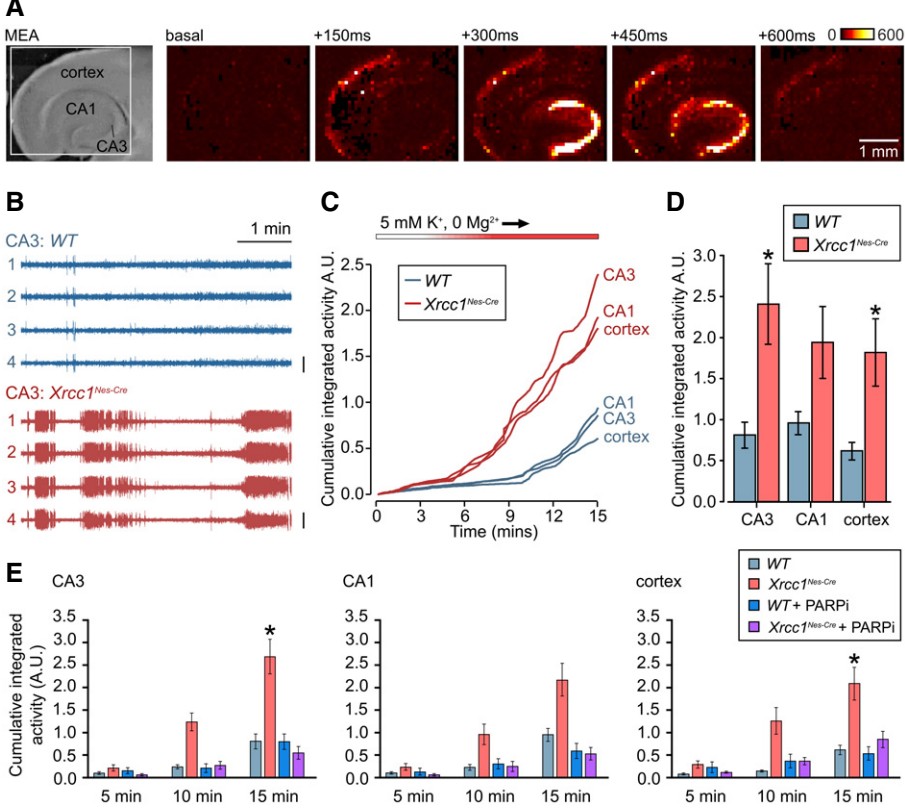

**Figure 4. Suppression of seizure-like activity in *Xrcc1^Nes-Cre* brain slices by PARP1 inhibitor.**

A  Brightfield image (left) of an acute brain slice positioned on HD-MEA with targeted regions indicated. (Right) progression of a typical seizure event in a slice perfused with epileptogenic buffer; colour code indicates voltage changes in microvolts.

B  Representative traces from four channels in the CA3 region of hippocampus recorded from 10 to 15 min in epileptogenic buffer, showing seizure-like activity in *Xrcc1^Nes-Cre*. Vertical scale bar indicates 300 μV.

C  Mean cumulative activity plots in CA1 and CA3 regions of hippocampus and cortex over 15 min of recording in epileptogenic buffer. *WT* (n = 8 slices from four mice), *Xrcc1^Nes-Cre* (n = 9 slices from four mice).

D  Summary histograms of the mean (± SEM) cumulative seizure-like activity at 15 min, from the data in panel C. Statistical significance was assessed by ANOVA with *post hoc* pairwise comparisons (*P < 0.05).

E  Summary histograms (mean ± SEM) of cumulative seizure-like activity at 5-, 10- and 15-min timepoints in cortex, CA1 and CA3 regions of wild-type and *Xrcc1^Nes-Cre* brain slices from mice treated or not for 5–8 days *ad libitum* with PARP1 inhibitor (ABT-888) prior to analysis. *WT* (n = 8 slices from four mice), *Xrcc1^Nes-Cre* (n = 10 from five mice), *WT* + PARPi (n = 6 from three mice) and *Xrcc1^Nes-Cre* + PARPi (n = 9 from three mice). Pairwise comparisons at the 15-min timepoint between WT versus *Xrcc1^Nes-Cre* mice and WT versus *Xrcc1^Nes-Cre* mice + PARPi were conducted by Kruskal–Wallis with Dunn's *post hoc* tests, and statistically significant differences (*P < 0.05) are shown.

(PARGi), the enzyme primarily responsible for poly(ADP-ribose) catabolism (Fig 5A and B). To confirm that the elevated poly(ADP-ribose) detected here was nascent polymer resulting from hyperactive Parp1, rather than pre-existing poly(ADP-ribose), we co-incubated the neurons with an inhibitor of PARP1 (PARPi). Indeed, the presence of PARPi ablated the appearance of poly(ADP-ribose) in the *Xrcc1^Nes-Cre* neurons (Fig 5A and B). This result demonstrates that Parp1 hyperactivation occurs continuously in *Xrcc1^Nes-Cre* hippocampal neurons, presumably as a result of the elevated steady-state level of unrepaired SSBs.

Next, to measure presynaptic calcium signalling, we transduced dissociated hippocampal cultures from different genotypes with SyGCaMP6f, a presynaptically targeted optical $Ca^{2+}$ reporter (Fig 6A and B) (Dreosti *et al*, 2009). We then carried out time-lapse imaging at DIV15–17 to assess $Ca^{2+}$ dynamics in response to electrical

stimulation. We found that with repeated presentations of 10 Hz stimulus trains, SyGCaMP6f-positive puncta in wild-type mouse cultures showed characteristic transient increases in fluorescence consistent with activity-evoked $Ca^{2+}$ influx at the presynaptic terminal (Fig 6C). Strikingly, however, the amplitude of these responses was ~ 2-fold higher in *Xrcc1^Nes-Cre* neurons, indicating that Xrcc1 loss results in excessive activity-evoked synaptic $Ca^{2+}$ influx (Fig 6C and D). Moreover, this defect was partially or fully suppressed by deletion of one or both alleles of *Parp1*, respectively, suggesting that the excessive activity-evoked synaptic $Ca^{2+}$ influx was a result of Parp1 hyperactivity (Fig 6C, E and G). To confirm this, we incubated cultures with PARP1 inhibitor (PARPi) continuously for 9–11 days prior to recording. Strikingly, this treatment fully suppressed the aberrant $Ca^{2+}$ response in *Xrcc1^Nes-Cre* neurons (Fig 6C, F and G). To our knowledge, this is the first demonstration

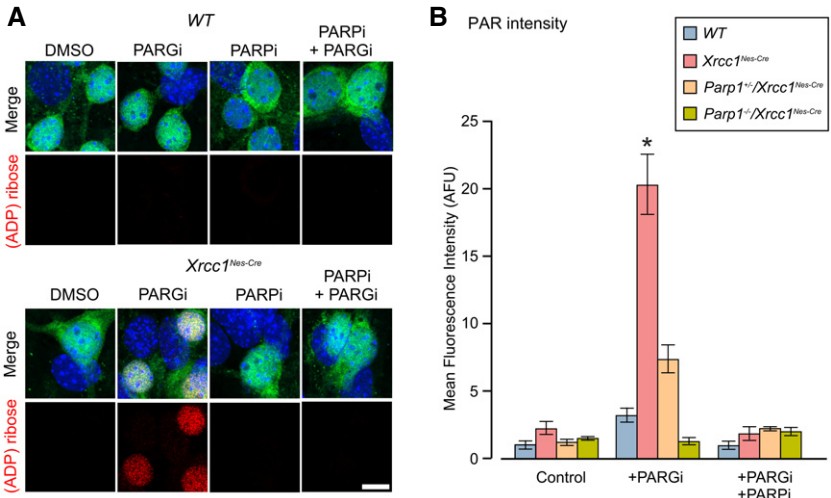

**Figure 5. Parp1 hyperactivation in isolated $Xrcc1^{Nes-Cre}$ hippocampal neurons.**

A  Representative images of indirect immunofluorescence of DIV6 hippocampal neurons cultured from P1 WT and $Xrcc1^{Nes-Cre}$ mouse pups, immunostained for ADP-ribose (red), NeuN to identify neurons (green) and counterstained with DAPI (blue). Cells were pretreated with PARP inhibitor (10 μM) or vehicle for 2 h prior to fixation, with PARG inhibitor (10 μM) additionally present for the final hour. Scale bar 10 μm.

B  Histogram of mean (± SEM) relative pan-ADP-ribose fluorescence in NeuN-positive hippocampal neurons pretreated with PARP inhibitor (5 μM) or vehicle for 5 h prior to fixation, with PARG inhibitor (10 μM) additionally present for the final hour. Neurons were cultured from WT (n = 6 mice, > 180 cells per condition), $Xrcc1^{Nes-Cre}$ (n = 6, > 180), $Parp1^{+/-}/Xrcc1^{Nes-Cre}$ (n = 3, > 90) and $Parp1^{-/-}/Xrcc1^{Nes-Cre}$ (n = 3, > 90). * indicates significant differences from WT (Kruskal–Wallis ANOVA, P = 0.0013 and Dunn's post hoc tests).

that aberrant Parp1 activity deregulates synaptic $Ca^{2+}$ signalling, providing a compelling explanation for the elevated seizures and, consequently, shortened lifespan in $Xrcc1^{Nes-Cre}$ mice.

## Discussion

DNA single-strand breaks (SSBs) are the commonest DNA lesions arising in cells and can block the progression of DNA and RNA polymerases (Hsiang et al, 1989; Zhou & Doetsch, 1993, 1994; Tsao et al, 1993; Kathe et al, 2004; Caldecott, 2008; Neil et al, 2012). The collision of DNA polymerases with SSBs can also result in DNA replication fork collapse and the formation of DSBs (Ryan et al, 1991; Strumberg et al, 2000; Kuzminov, 2001). However, proliferating cells possess effective and accurate homologous recombination mechanisms by which replication-associated DSBs can be repaired using an intact sister chromatid (Haber, 1999; Arnaudeau et al, 2001; Costes & Lambert, 2012), perhaps explaining why human diseases in which SSBR is attenuated do not result in markedly elevated genome instability and cancer (Caldecott, 2008; Yoon & Caldecott, 2018). Consistent with this idea, proliferating cells from individuals with genetic defects in SSBR possess elevated levels of sister chromatid exchange, a hallmark of homologous sister chromatid recombination (El-Khamisy et al, 2005; Hoch et al, 2017).

In contrast to proliferating cells, post-mitotic cells lack sister chromatid recombination and so are more reliant on SSBR, perhaps explaining why defects in the latter pathway are primarily associated with neurological dysfunction (McKinnon & Caldecott, 2007; Caldecott, 2008; Yoon & Caldecott, 2018). In addition, SSBs may arise in neurons at higher frequencies than other cell types, for example as a result of glutamate excitotoxicity (Mandir et al, 2000)

and/or as a result of processes associated with gene transcription. As an example of the latter, topoisomerase I activity can induce protein-linked SSBs during gene transcription and these have been implicated previously in SSBR-defective neurodegenerative diseases (Takashima et al, 2002; El-Khamisy et al, 2005; Katyal et al, 2014; Kalasova et al, 2020). In addition, the modified base 5'-hydroxymethylcytosine is highly enriched in brain and may generate SSBs as an obligate intermediate of DNA base excision repair, during epigenetic reprogramming (Kriaucionis & Heintz, 2009; Li & Liu, 2011).

Arguably the most severe pathology observed in SSBR-defective disease is neurological seizures. However, the molecular mechanisms by which unrepaired SSBs trigger these potentially lethal events are unknown. Here, we have identified one such mechanism. We found that genetic deletion of *Parp1* greatly suppressed the increased seizures in $Xrcc1^{Nes-Cre}$ mice, demonstrating for the first time a causal impact of excessive/aberrant Parp1 activity on these events. Consistent with this, we found that poly(ADP-ribose) levels are elevated across $Xrcc1^{Nes-Cre}$ brain and are particularly high in the hippocampus, a region of the brain commonly associated with seizure activity. We also detected elevated levels of poly(ADP-ribose) in dissociated $Xrcc1^{Nes-Cre}$ hippocampal neurons, which we confirmed was the result of ongoing Parp activity. The presence of elevated poly(ADP-ribose) is consistent with the SSBR defect in $Xrcc1^{Nes-Cre}$ mice, because the synthesis of this polymer is triggered by endogenous SSBs (Benjamin & Gill, 1980; Ikejima et al, 1990; Eustermann et al, 2015; Hanzlikova et al, 2018). Nevertheless, that the level of endogenous poly(ADP-ribose) was high enough to detect in $Xrcc1^{Nes-Cre}$ brain slices was surprising, and is consistent with the idea discussed above that SSB levels are particularly elevated in brain.

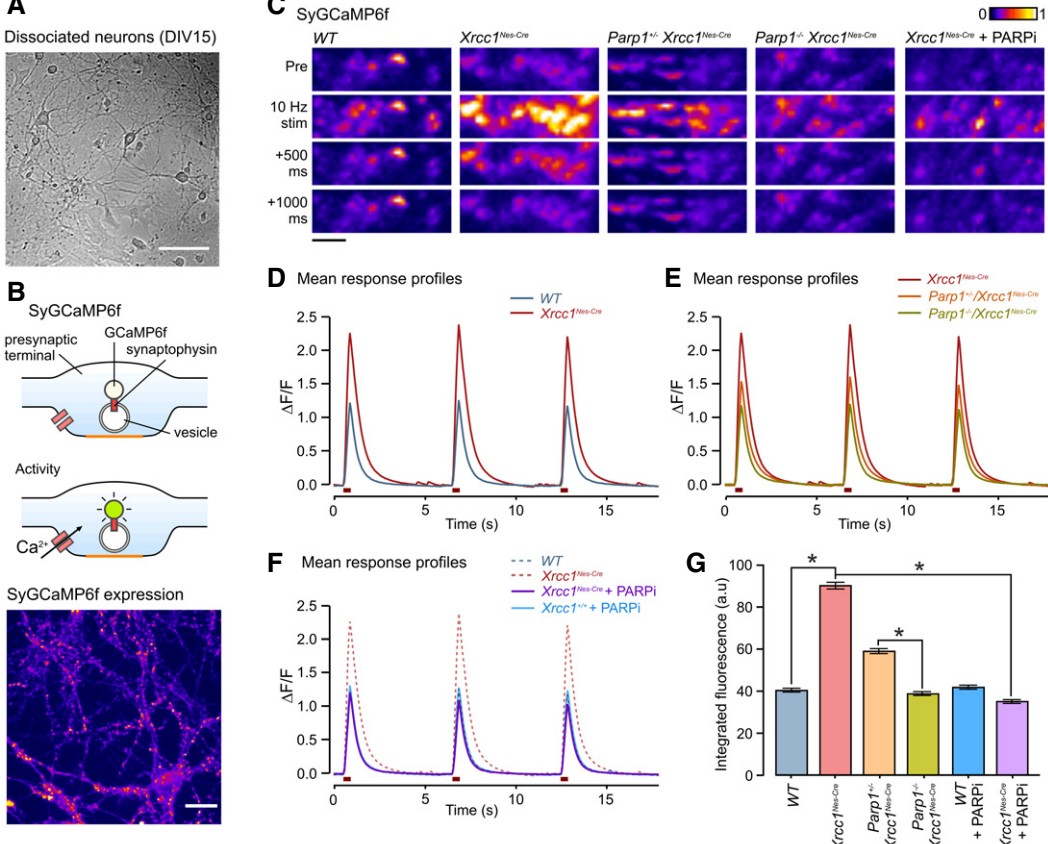

**Figure 6. Aberrant presynaptic calcium signalling in *Xrcc1*$^{-/-}$ hippocampal neurons is rescued by Parp1 deletion or inhibition.**

A       Representative image of DIV15 dissociated cultured neurons used. Scale bar: 100 μm.

B       (Top) Cartoon schematic illustrates SyGCaMP6f targeting and action. (Bottom) Image shows typical punctate SyGCaMP6f expression at DIV15. Scale bar, 20 μm.

C       Representative images of fluorescence responses in synaptic terminals expressing SyGCaMP6f to a train of 10 stimuli at 10 Hz in dissociated hippocampal neurons derived from wild-type and mutant mice. Scale bar: 5 μm.

D, E    Mean SyGCaMP6f responses to three rounds of 10 action potentials stimulation from mice of the following genotypes: *WT* (n = 1,946 synapses, nine coverslips, three animals), *Xrcc1$^{Nes-Cre}$* (n = 3,313, 12, 4), *Parp1$^{+/-}$/Xrcc1$^{Nes-Cre}$* (n = 2,272, 9, 3) and *Parp1$^{-/-}$/Xrcc1$^{Nes-Cre}$* (n = 2,122, 11, 3).

F       Response to chronic treatment with the PARP inhibitor, KU 0058948 Hydrochloride (1 μM) for 9–11 days prior to imaging for *Xrcc1$^{+/+}$* (n = 1,264 synapses, eight coverslips, three animals) and *Xrcc1$^{Nes-Cre}$* (n = 2,231, 10, 4) mice.

G       Summary histogram of the mean (± SEM) of the integrated fluorescence responses for all individual synapses for each condition. Synapse number is the same as in (D–F). Responses are significantly higher in *Xrcc1$^{Nes-Cre}$* synapses versus both *WT* and *Parp1$^{-/-}$/Xrcc1$^{Nes-Cre}$* (one-way ANOVA, *$P < 0.0033$ and pairwise Student's *t*-tests).

Data information: (D–F) Data are from three or more independent experiments per genotype/treatment type, combined into one experimental data set and plotted on three separate graphs for clarity. Dashed lines in panels E and F are the *WT and/or Xrcc1$^{Nes-Cre}$* curves transposed from panel D for comparative purposes.

To our knowledge, these data are the first to demonstrate a molecular mechanism by which unrepaired DNA strand breaks trigger neurological seizures. Seizures are potentially lethal events, associated with a condition denoted sudden unexpected death during epilepsy (SUDEP) (Smithson *et al*, 2014; Buchhalter & Cascino, 2017). Indeed, *Xrcc1$^{Nes-Cre}$* mice exhibit a dramatically shorted lifespan, which by video imaging we established here is due to episodic epilepsy leading ultimately to a fatal seizure. The cause of death in *Xrcc1$^{Nes-Cre}$* mice during seizure is unclear but, similar to SUDEP, is likely to result from the disruption of normal cardiac or respiratory function (Glasscock, 2014). The discovery that deletion of one *Parp1* allele prolonged the lifespan of *Xrcc1$^{Nes-Cre}$* mice to a greater extent (~25-fold) than did deletion of both *Parp1* alleles (~ 7-fold) is surprising. This indicates that whilst loss of one allele of

*Parp1* is sufficient to suppress Parp1-induced lethality, additional loss of the second allele eradicates an as yet unidentified role for Parp1 in Xrcc1-defective brain that is important for long-term survival. This role likely involves the signalling and/or processing of SSBs, because it is required for longevity only in the absence of Xrcc1. The cause of the shortened lifespan in *Xrcc1$^{Nes-Cre}$* mice lacking both *Parp1* alleles is unclear, but is unlikely to involve seizures because we failed to detect elevated seizure-like activity in brain sections from these mice, *ex vivo*.

PARP1 hyperactivity can trigger cell death via excessive depletion of NAD$^{+}$, inhibition of glycolysis, and/or by a specialized type of apoptosis known as parthanatos (Zhang *et al*, 1994; Eliasson *et al*, 1997; Andrabi *et al*, 2006; Yu *et al*, 2006; Aredia & Scovassi, 2014; Fouquerel *et al*, 2014). However, our discovery that aberrant

Parp1 activity resulted in aberrant $Ca^{2+}$ signalling at presynaptic terminals in viable neurons in culture suggests that Parp1 impacts on synaptic function, directly. It will now be important to identify how Parp1 hyperactivity leads to deregulated calcium homeostasis and to understand the downstream consequences of this deregulation on the propagation of information through neural circuits. It is possible that Parp1 hyperactivity affects $Ca^{2+}$ signalling, in part at least, as a result of $NAD^+$ depletion. Consistent with this possibility, we discovered in the current work that $NAD^+$ levels are reduced by ~ 50% in Xrcc1$^{Nes-Cre}$ brain. This is because this enzyme co-factor is required for the synthesis of several second messengers involved in calcium mobilization and release (Guse, 2015). In addition, reduced levels of $NAD^+$ can affect the expression of genes encoding regulators of calcium homeostasis, as has been reported in the neurodegenerative disease spinocerebellar ataxia type 7 (Stoyas *et al*, 2019). Alternatively, or in addition, Parp1 hyperactivity might deregulate calcium signalling more directly, via inappropriate ADP-ribosylation of itself or other proteins that regulate gene expression, for example.

It is worth noting that the defects reported here in Parp1 hyperactivity, $NAD^+$ levels and calcium signalling likely extend beyond the hippocampus and seizure phenotypes to other neurological pathologies. Indeed, we have shown previously that Parp1 hyperactivation accounts for the cerebellar dysfunction and ataxia that is observed in Xrcc1$^{Nes-Cre}$ mice (Hoch *et al*, 2017). It will now be of interest to investigate the impact of $NAD^+$ depletion and/or deregulated calcium homeostasis on these pathologies too. In addition, although speculative, our data may also be relevant to other ageing-related neurodegenerative conditions, and perhaps even normal human ageing, in which altered levels of PARP1 activity, $NAD^+$ metabolism and calcium homeostasis are features (Maynard *et al*, 2015; Stoyas *et al*, 2019).

Finally, it is also noteworthy that we were able to ablate the elevated seizure-like activity in Xrcc1-defective hippocampal slices by chronic application of PARP inhibitor in the drinking water of the mother. This is an exciting finding, because it supports the possibility that PARP inhibition might provide a therapeutic approach for the treatment of XRCC1-defective, and possibly other, neurological diseases. It should be noted however that currently available inhibitors may not be suitable for this purpose, however, because they "trap" PARP enzymes on unrepaired SSBs and so exacerbate DNA repair defects, thereby increasing DNA replication fork stalling and/or collapse during S phase. Whilst this is not a problem for post-mitotic neurons, proliferating XRCC1-defective cells are hypersensitive to inhibitors that trap PARP1 (Ali *et al*, 2018, 2020). Although the PARP inhibitor employed here for our *in vivo* experiments (ABT-888; Veliparib) is a relatively weak "trapper" (Murai *et al*, 2012), this may explain why we have so far been unable to extend significantly the lifespan of XRCC1$^{Nes-Cre}$ mice by PARP inhibition.

In summary, we reveal here that aberrant Parp1 activity triggers seizures and shortened lifespan in Xrcc1$^{Nes-Cre}$ mice. We demonstrate seizure-like activity in Xrcc1-defective hippocampus *ex vivo* and aberrant presynaptic $Ca^{2+}$ signalling in hippocampal neurons *in vitro*, and we show that these defects are suppressed by Parp1 deletion or inhibition. These data highlight Parp1 inhibition as a possible therapeutic approach for the treatment of XRCC1-defective neurological disease.

# Materials and Methods

## Animals and animal care

Experiments were carried out in accordance with the UK Animal (Scientific Procedures) Act 1986 and satisfied local institutional regulations at the University of Sussex. Mice were maintained and used under the auspices of UK Home Office project licence number P3CDBCBA8. The generation of Parp1$^{-/-}$, Xrcc1$^{Nes-Cre}$ and Ku70$^{-/-}$ mice was reported previously (Wang *et al*, 1995; Lee *et al*, 2009). Intercrosses between Parp1$^{-/-}$ and Xrcc1$^{+/loxp}$ mice were maintained in a mixed background C57Bl/6 × S129 strain and housed on a 12-h light/dark cycle with lights on at 07:00. Temperature and humidity were maintained at 21°C (± 2°C) and 50% (± 10%), respectively. All experiments were performed under the UK Animal (Experimental Procedures) Act, 1986.

## Antibodies

Antibodies used were mouse monoclonal anti-PARP1 (Serotec; MCA1522G), rabbit Fc-fused anti-poly-ADP-ribose binding reagent (Millipore; MABE1031), rabbit Fc-fused anti-pan-ADP-ribose binding reagent (Millipore; MABE1016), rabbit polyclonal anti-poly-ADP-ribose (Trevigen; 4336), rabbit monoclonal anti-ATM [EPR17059] (Abcam; ab199726), mouse monoclonal anti-NeuN (A60, Millipore; MAB337), rabbit polyclonal anti-XRCC1 (Novus Biologicals; NBP1-87154) and rat polyclonal anti-α-tubulin (YL1/2, Abcam; ab6160). Secondary antibodies used for immunofluorescence were goat anti-rabbit Alexa 647 and anti-mouse Alexa 488 (Invitrogen; A21244 and A11001) and for immunohistochemistry Biotin-SP-conjugated Affini-Pure goat anti-rabbit antibody (Jackson ImmunoResearch; 111-065-144).

## Immunohistochemistry and microscopy

Mice were anaesthetized using 0.25 mg/g Dolethal (Vetoquinol UK Ltd) and perfused transcardially with PBS followed by 4% formaldehyde. Brains were postfixed in 4% paraformaldehyde for 48 h and stored in 25% sucrose/PBS until moulding and freezing (TFM-5). 10–20 μm sagittal sections were prepared using a cryostat (Leica CM1850). Immunohistochemistry was conducted as described previously (Hoch *et al*, 2017). Fluorescent images were acquired using the Zeiss LSM880 confocal microscope, with oil immersion objective (Plan-Apochromat 63×/1.4 Oil DIC M27). The Airyscan super-resolution module was used to obtain high-resolution images. The image stacks were processed for brightness and contrast in ImageJ 1.51j. Images of immunohistochemistry samples were acquired with LSM880 using the wide-field imaging mode (Axiocam 503 Mono, Plan-Apochromat 10×/0.45 and 20×/0.8 M27). Nuclear staining was quantified manually using ImageJ 1.51j. The mean nuclear signal was normalized by subtracting the mean value for background staining. Whole tissue sections were imaged in a tiling mode with 10% overlap and stitched in Zeiss Zen.

## ADP-ribosylation assay and Western blotting

Mouse brains were dissected from P15 mouse pups following cervical dislocation. The cerebellum and the forebrain containing cortex and hippocampus were isolated separately. The frozen tissues were

processed in glass Dounce homogenizer on ice in lysis buffer containing 20 mM HEPES pH 7.4, 2 mM EGTA, 1 mM DTT, 1 % v/v Triton X-100, 10% v/v glycerol, 1× protease inhibitor cocktail (Roche cOmplete, EDTA-free) and 1% v/v phosphatase inhibitor cocktail (Sigma 003). After 15 min on ice with occasional swirling, unbroken cells were removed at 500 $g$ in a cooled centrifuge for 10 min. The salt concentration of the supernatant was adjusted to 100 mM KCl, 2.5 mM $MgCl_2$, and protein concentration determined using the BCA assay (Pierce). For the ADP-ribosylation assay, DMSO or 10μM PARPi (KU0058948 hydrochloride, Axon) as indicated was combined with the tissue lysates 15 min prior to initiation of the reaction by addition of 1 mM β-Nicotinamide adenine dinucleotide ($NAD^+$, NEB). Aliquots of each reaction were stopped after 5, 10, 15, 30 and 45 min in 5× Laemmli buffer (2.5 mM Tris–HCl pH 6.8, 0.5 mM DTT, 1% SDS, 50% glycerol, 0.05% Bromophenol blue). The samples were subjected to SDS–PAGE (7 or 10%), proteins transferred onto nitrocellulose membrane and detected using the poly(ADP-ribose)-specific detection reagent MABE1031, combined with horseradish peroxidase-conjugated secondary antibodies. Peroxidase activity was detected by ECL reagent (Clarity™ Western ECL Substrate, Bio-Rad) and Medical X-ray Film Blue (Agfa HealthCare).

## NAD$^+$ assay

$NAD^+$ levels in mice brain tissue were determined by a chromogenic assay as described before (Baker *et al*, 2016). Briefly, forebrain tissue containing cortex and hippocampus was dissected from P15 mice. The tissue was washed in PBS and homogenized in lysis buffer [20 mM sodium bicarbonate, 100 mM sodium carbonate, 0.5% Triton X-100, 10 mM nicotinamide, 100 μM PARG inhibitor (PDD00017273, Sigma), 40 μM PARP inhibitor (KU0058948, Axon) and cOmplete protease inhibitors (04693132001, Roche), pH 10.3] using TissueLyser II (Qiagen). The cells were further lysed by two freeze/thaw cycles. Proteins concentrations were measured in samples after centrifugation and normalized, and the lysates then added to a 10,000 MWCO centrifugal filter (UFC501024, Merck™) and centrifuged at 14,000 $g$ at 4°C for 30 min. Half of each lysate was incubated at 60°C for 30 min to decompose $NAD^+$. Samples were incubated in cycling buffer [100 mM tricine-NaOH (pH 8), 4 mM EDTA, 40 mM NaCl, 1.66 mM phenazine ethosulfate (PES), 0.42 mM thiazolyl blue tetrazolium bromide (MTT), 10% ethanol] at 37°C for 5 min, and 10 U/ml alcohol dehydrogenase, reconstituted in 100 mM tricine-NaOH (pH 8), was added to drive a cycling reaction at 37°C for 40 min. The reaction was terminated by addition of NaCl (2 M final concentration), and samples were centrifuged at 10,000 $g$ at 4°C for 5 min. Reduced MTT was resuspended in 100% ethanol, and the absorbance was measured at 560 nm. $NAD^+$ concentrations were calculated by subtracting the NADH concentration in the sample in which $NAD^+$ was decomposed prior to the cycling reaction from that which was not.

## Electrophysiology

For targeted extracellular recordings, acute transverse hippocampal slices (300 μm) were prepared from P14-P16 mice using a vibroslicer (VT1200S, Leica Microsystems, Germany) in ice-cold artificial cerebrospinal fluid containing (in mM): 125 NaCl, 2.5 KCl, 25 glucose, 1.25 $NaH_2PO_4$, 26 $NaHCO_3$, 1 $MgCl_2$, 2 $CaCl_2$ (bubbled with 95% $O_2$ and 5% $CO_2$, pH 7.3). All experiments were performed at 23–25°C. During an experiment, an extracellular electrode was placed in the hippocampal CA3 pyramidal region and field voltage recordings made as slices were perfused from ACSF into a modified (epileptogenic) saline containing (in mM) 125 NaCl, 5 KCl, 25 glucose, 1.25 $NaH_2PO_4$, 26 $NaHCO_3$, 2 $CaCl_2$. Signals were amplified using a MultiClamp 700A (Molecular Devices), digitized at 50 kHz with a Digidata 1320 and recorded in pCLAMP acquisition software (Molecular Devices) for offline analysis. To quantify seizure-like activity, raw traces were exported and analysed in MATLAB (MathWorks). Root mean square signal envelopes (sliding window length: 400 samples) were calculated for each trace and a peak waveform generated using an automated peak-find search function. This waveform was then enveloped (sliding window length: 1,500 samples) and a peak count analysis used to quantify seizure-like episodes (SLEs) for each sample. To independently verify our automated analysis approach, we also carried out a separate manual count of SLEs, by tallying episodes in 2.5 s time bins. Outputs from both types of quantification were highly significantly positively correlated (Spearman rank, $r_s = 0.833$, $P < 0.0001$).

For MEA recordings, slices were placed onto a high-density Stimulo MEA chip (4096 electrodes: size 21 × 21 μm, pitch 81 μm, 64 × 64 matrix, 3Brain). The slices were immobilized using a custom-made weight under membrane and were constantly perfused with oxygenated (epileptogenic) ACSF (as above) at + 34°C. Recordings were acquired with BrainWave v4.2 software in 10-min time windows, digitized at 9.5 kHz and stored for offline analysis. 7-channel arrays from target regions were batch-exported in 3Brain HDF5 format and quantified in MATLAB. Signals were processed first by enveloping (RMS signal envelope, sliding window length: 500 samples) and then by generating a peak waveform to identify seizure-like events. These waveforms were then integrated to provide a collective measure of burst frequency and amplitude at each timepoint and presented as cumulative totals. Where indicated, the PARP inhibitor ABT-888 Hydrochloride (Selleckchem) was administered in drinking water (250 μg/ml, *ad libitum)* to the mouse dam and pre-weaned pups from their post-natal day 10 (P10) until analysis at P14–17.

## Video analysis

Video monitoring was performed using the Noldus PhenoTyper 3000 system, including infrared LED units and a video recording camera for the duration of the experiment [video output CCIR black/white VPP −75 Ohm (PAL) or EIA black/white Vpp-75 Ohm (NTSC)]. Mice were placed in the chamber with floor area 30 cm × 30 cm and were provided with bedding, minimal nesting, food pellets and a water source. Mouse pups of the indicated genotype were housed with mother and a control sibling from P15 up to P20. Video recordings were observed after recording, and the number of running–bouncing seizures was quantified.

## Cell culture

P1–2 mouse pups were decapitated, and brains were removed and placed in ice-cold Hank's buffered saline solution (HBSS), 0.1 M HEPES. Hippocampi were isolated and then washed three times in

warmed 10% FCS, 20 mM glucose and 1% Pen/Strep Minimal Essential Media (MEM) (Gibco). Hippocampi were manually dissociated by trituration and diluted to a density of 50,000 cells/well and plated on poly-D-Lysine (20 µg/ml) coated 15-mm glass coverslips. 2 h after plating, the media was replaced with 2% B27, 1% glutamax and 1% Pen/Strep-supplemented neurobasal A medium (Gibco). Cells were maintained without cytosine arabinoside, typically used to manage glial cell count, to prevent induction of exogenous DNA damage (Zhuo *et al*, 2018). Cells were maintained in 5% $CO_2$ at 37°C and fed every 3 days through half-exchange of media. Hippocampal cells were taken for ICC experiments at DIV6 and live imaging at DIV15–17. Hippocampal cells treated for PAN immunofluorescence were administered 10 µM PARG inhibitor PDD 00017273 (Tocris) in DMSO for 1 h following pre-treatment with 5 µM PARPi Ku-0058948 or DMSO vehicle. Cells were fixed in 4% paraformaldehyde and permeabilized with 0.2% Triton X-100 for 2 min. Cells were blocked in 10% goat serum and 0.3% Triton X-100 for 30 min prior to incubation with relevant primary antibodies. Cells were incubated with relevant secondary antibodies following PBST washes and counterstained with DAPI prior to mounting.

### SyGCaMP6f imaging

Hippocampal neurons were transduced with AAV6_SyGCaMP6f at DIV6/7 at a multiplicity of infection (MOI) of 100. Images were acquired using a 60×/1.0 NA objective on an Olympus BX61WI microscope fitted with an Andor Ixon + EM-CCD (40 ms exposure, 20 Hz acquisition frequency and 4 × 4 binning) controlled by custom-written Micromanager routines. Coverslips were placed in a custom-built imaging chamber, and a Grass SD9 Stimulator was used to apply field stimulation (voltage: 22.5V, 1 ms pulse width). All experiments were carried out in Extracellular Bath Solution (EBS) containing (in mM), 136 NaCl, 10 HEPES, 10 D-Glucose, 2.5 KCl, 2 $CaCl_2$ and 1.3 $MgCl_2$. EBS was supplemented with 50 µM APV and 20 µM CNQX to inhibit NMDA and AMPA receptors, respectively. Image stacks were analysed using IGOR Pro 8. Stacks were imported via the SARFIA plugin (Dorostkar *et al*, 2010) and corrected for x-y drift using the built-in image registration function (Thevenaz & Unser, 1998). Regions of Interest (ROIs) were detected via image segmentation using a threshold of $(-3) \times$ SD of all pixel values in the Laplace operator. Mean background intensity was subtracted to account for alterations to background fluorescence, and $\Delta F/F$ values were calculated using the 10 frames prior to stimulation as a baseline. A custom script was used to determine whether ROIs had exceeded a threshold change in fluorescence intensity following stimulation and ROIs that failed to reach threshold intensity were removed from the mask.

## Data availability

No large-scale datasets are associated with this work. All raw data and materials associated with the figures are available on request.

Expanded View for this article is available online.

## Acknowledgements
We thank Zhao-Qi Wang for the *Parp1*$^{-/-}$ mouse strain and Tom Baden and Marvin Seifert for their support with the MEA work. This work was funded by an MRC Programme Grant to KWC and KS (MR/P010121/1), an ERC Advanced Investigator Award to KWC (SIDSCA; 694996) and BBSRC Project grants to KS (BB/K019015/1; BB/S00310X/1). KWC is the recipient of a Royal Society Wolfson Research Merit Award.

## Author contributions
KWC and KS conceived and designed the study. EK and SRu conducted the lifespan experiments. EK conducted IHC, MEA, electrophysiology and video analysis, with help from SRe. EK conceived and JB conducted the calcium and neuron experiments, with help from KF. IK and KI conducted the $NAD^+$ measurements, with supervision by HH. PJM provided the Xrcc1 mouse model, and LJ managed the mouse colonies. KWC and KS wrote the manuscript, with editing from HH, EK and JB.

## Conflict of interest
The authors declare that they have no conflict of interest.

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
