## [Review Process File · EMBO Reports]

Parp1 Hyperactivity Couples DNA Breaks to Aberrant Neuronal Calcium Signalling and Lethal Seizures

Emilia Komulainen, Jack Badman, Stephanie Rey, Stuart Rulten, Limei Ju, Kate Fennell, Ilona Kalasova, Kristyna Ilievova, Peter McKinnon, Hana Hanzlikova, Kevin Staras, and Keith Caldecott
DOI: 10.15252/embr.202051851

Corresponding author(s): Keith Caldecott (k.w.caldecott@sussex.ac.uk) , Kevin Staras (k.staras@sussex.ac.uk)

Review Timeline:

Submission Date:	6th Oct 20
Editorial Decision:	11th Nov 20
Revision Received:	8th Feb 21
Editorial Decision:	1st Mar 21
Revision Received:	4th Mar 21
Accepted:	5th Mar 21

Editor: Esther Schnapp

Transaction Report:

Dear Keith,

Thank you for the submission of your manuscript to EMBO reports. We have now received the full set of referee reports that is pasted below.

As you will see, all referees acknowledge that the findings are interesting. However, they also suggest some more experiments to strengthen the study. I think that all points raised are interesting and should be addressed, but please let me know in case you disagree and we can discuss the revisions further, also per video chat, if this is easier for you.

I would thus like to invite you to revise your manuscript with the understanding that the referee concerns must be fully addressed and their suggestions taken on board. Please address all referee concerns in a complete point-by-point response. Acceptance of the manuscript will depend on a positive outcome of a second round of review. It is EMBO reports policy to allow a single round of major revision only and acceptance or rejection of the manuscript will therefore depend on the completeness of your responses included in the next, final version of the manuscript.

Revised manuscripts should be submitted within three months of a request for revision; they will otherwise be treated as new submissions. Please contact us if a 3-months time frame is not sufficient for the revisions so that we can discuss this further.

Regarding data quantification, please specify the number "n" for how many independent experiments were performed, the bars and error bars (e.g. SEM, SD) and the test used to calculate p-values in the respective figure legends. This information must be provided in the figure legends. Please also include scale bars in all microscopy images.

- 1) A data availability section providing access to data deposited in public databases is missing. If you have not deposited any data, please add a sentence to the data availability section that explains that.
- 2) Your manuscript contains statistics and error bars based on $n=2$. Please use scatter blots in these cases. No statistics should be calculated if $n=2$.

- 2) individual production quality figure files as .eps, .tif, .jpg (one file per figure).

See https://wol-prod-cdn.literatumonline.com/pb-assets/embo-site/EMBOPress_Figure_Guidelines_061115-1561436025777.pdf for more info on how to prepare your figures.

- 3) We replaced Supplementary Information with Expanded View (EV) Figures and Tables that are collapsible/expandable online. A maximum of 5 EV Figures can be typeset. EV Figures should be cited as 'Figure EV1, Figure EV2' etc... in the text and their respective legends should be included in

the main text after the legends of regular figures.

5) a complete author checklist, which you can download from our author guidelines <<https://www.embopress.org/page/journal/14693178/authorguide>>. Please insert information in the checklist that is also reflected in the manuscript. The completed author checklist will also be part of the RPF.

6) Please note that all corresponding authors are required to supply an ORCID ID for their name upon submission of a revised manuscript (<<https://orcid.org/>>). Please find instructions on how to link your ORCID ID to your account in our manuscript tracking system in our Author guidelines <<https://www.embopress.org/page/journal/14693178/authorguide#authorshipguidelines>>

7) Before submitting your revision, primary datasets produced in this study need to be deposited in an appropriate public database (see <https://www.embopress.org/page/journal/14693178/authorguide#datadeposition>). Please remember to provide a reviewer password if the datasets are not yet public. The accession numbers and database should be listed in a formal "Data Availability" section placed after Materials & Method (see also <https://www.embopress.org/page/journal/14693178/authorguide#datadeposition>). Please note that the Data Availability Section is restricted to new primary data that are part of this study. * Note - All links should resolve to a page where the data can be accessed. *
If your study has not produced novel datasets, please mention this fact in the Data Availability Section.

8) We would also encourage you to include the source data for figure panels that show essential data. Numerical data should be provided as individual .xls or .csv files (including a tab describing the data). For blots or microscopy, uncropped images should be submitted (using a zip archive if multiple images need to be supplied for one panel). Additional information on source data and instruction on how to label the files are available at <<https://www.embopress.org/page/journal/14693178/authorguide#sourcedata>>.

9) Our journal also encourages inclusion of *data citations in the reference list* to directly cite datasets that were re-used and obtained from public databases. Data citations in the article text are distinct from normal bibliographical citations and should directly link to the database records from which the data can be accessed. In the main text, data citations are formatted as follows: "Data ref: Smith et al, 2001" or "Data ref: NCBI Sequence Read Archive PRJNA342805, 2017". In the

Reference list, data citations must be labeled with "[DATASET]". A data reference must provide the database name, accession number/identifiers and a resolvable link to the landing page from which the data can be accessed at the end of the reference. Further instructions are available at <https://www.embopress.org/page/journal/14693178/authorguide#referencesformat>

I look forward to seeing a revised version of your manuscript when it is ready. Please let me know if you have questions or comments regarding the revision.

Best wishes,
Esther

Referee #1:

The manuscript by Komulainen et al is an investigation of the role of Parp1 in promoting or alleviating the symptoms of conditional Xrcc1 deletion (Xrcc1Nes-Cre). The authors report that death is likely caused by lethal seizures leading to a dramatically shortened lifespan. They use electrophysiological and optical approaches to demonstrate that increased Parp1 activity (caused by Xrcc1 deficiency) triggers seizure-like activity in vivo, in slices and in isolated hippocampal neurons in vitro. Under these conditions, Parp1 inhibition and/or deletion in conditional Xrcc1 knockout mice suppresses seizures, restores electrophysiological activity and lengthens lifespan. The authors speculate as to how PARP inhibition might serve as a therapeutic approach to treatment of XRCC1-dependent neurological disease.

I found this to be an absolutely lovely paper. The model is well chosen, the problems addressed are significant and the experimental approaches are well thought out and well controlled. I was particularly impressed by the sophisticated activity measurements used - the MEAs and the modified GCaMP6 optical imaging. I literally have no changes to request (and I don't think I have said that in one my reviews in many years).

My only regret is that the authors focused all of their energies on the hippocampus and to a lesser extent on the cortex. This is logical and in the context of the large PARP1 increases in these areas, perfectly appropriate. But the human condition has a prominent cerebellar phenotype and it would

have been of considerable interest to repeat some key experiments, particularly the MEA recordings, on slice preparations from this region to determine if there were lesser but still significant network changes there. Any answer would be interesting, even a negative one. For EMBO Reports, however, these additional observations are not required, and their absence does not diminish my strong enthusiasm for seeing this paper published in close to its current form.

Referee #2:

In this manuscript, Komulainen et al. show that lethal seizures and shortened lifespan due to loss of *Xrcc1* in the nervous system can be rescued by deleting PARP1. The authors further show that deleting PARP1 can correct defects in presynaptic calcium signaling in *Xrcc1*-deficient neurons, and also decrease seizure like activity in *Xrcc1*-deficient hippocampal slices. The experiments are well designed and executed. Importantly, the reduction of seizure-like activity in *Xrcc1*Nes-Cre animals by PARP1 deletion is quite interesting and new. Addressing the comments below would further strengthen the points made in the manuscript:

1. Figure 1 shows that loss of PARP1 can cause a reduction in ADP-Ribose levels in *Xrcc1* KO mice. Taken together with the reduced PARP1 staining, the authors conclude that PARP1 is hyperactive in *Xrcc1* KO brains. However, the figure does not directly show that the remaining PARP1 protein in these brains is actually hyperactive. The authors say throughout the text and figures that PARP1 is aberrant/hyperactive in *Xrcc1* KO brains, but this is never directly demonstrated biochemically.
2. Related to the point above, the quantification in Figure 1 shows that the loss of PARP1 causes a general reduction in ADP-Ribose levels unrelated to the loss of *Xrcc1* (blue vs gray bars). This could account for the lack of detectable ADP-ribose staining instead of a specific PARP1 hyperactivation in *Xrcc1* KO brains. Both points could be addressed by assessing the biochemical activity of PARP1 from these tissues.
3. Several correlations described by the authors need further clarification. For instance, in Figure 2, the observation is made that the loss of a single PARP1 copy is protective. However, in Figure 1, poly-ADP ribose levels are significantly elevated under these conditions. On the other hand, whereas the loss of PARP1 alone does not seem to affect mortality, its loss in the context of *Xrcc1* seems toxic. Based on these results, the correlation between PARP1 levels, PARP1 activity, and mortality seem unclear. Finally, the loss of a single copy of PARP1 is sufficient to reduce seizures even though this PARP1 would seem to be hyperactive based on the authors interpretations of Figure 1.
4. The results in Figure 4 showing the reduction of seizures following the loss of PARP1 in *Xrcc1* KO mice is indeed interesting. Does the loss of PARP1 also affect the ability to induce seizures in a WT background? The same comment extends to the results shown in Figure 6. Does the loss of PARP1 affect calcium signaling in a WT background?
5. The results in Figure 5 further suggest the importance of directly assessing PARP1 activity in WT and *Xrcc1* KO neurons and tissues.

Referee #3:

Komulainen et al. investigated the mechanism linking defects in DNA SSB with neurological dysfunction. The work showed hyperactivity of *Parp1* in *XRCC1NES-CRE* mice results in lethal seizures and shortened lifespan. And both defects are prevented by *Parp1* inhibition. Overall, the work is of interest and well written, and highlights PARP inhibition as a possible therapeutic

approach in XRCC1-mutated neurological disease.

My comments are as follows:

1. The authors checked Parp1 protein level, as well as ADP-ribose in mice brains. It's better to show PARylation in neurons and tissues.
2. How Parp1 hyperactivity affects Ca²⁺ signaling, one possibility is that this is a result of NAD⁺ depletion. The authors should measure NAD⁺ levels in the cells and tissues.
3. The lifespan and seizure results are good. Did you perform other behavior assays on the mice? Such as cognition and motor function assays, which are closely related to neurological diseases.
4. Besides Parp1, it's better to measure other DNA repair related proteins in XRCC1NES-CRE and Parp1^{-/-} XRCC1NES-CRE mice brains.
5. Parp1 inhibition seems to be a potential therapeutic target for XRCC1-mutated neurological disease. Then if treated XRCC1NES-CRE mice with Parp1 inhibitor, will the longevity and seizure be improved?
6. PARP1 and NAD⁺ axis in ageing-related disease and neurodegenerative diseases should be discussed, there are several relevant papers and reviews in the literature.

Dear Esther,

Please find below the point-by-point response to the referees of our manuscript. We would like to take this opportunity to thank the referees for their insightful comments, which I believe we have addressed in full and which have significantly improved the manuscript.

Referee #1:

The manuscript by Komulainen et al is an investigation of the role of Parp1 in promoting or alleviating the symptoms of conditional Xrcc1 deletion (Xrcc1Nes-Cre). The authors report that death is likely caused by lethal seizures leading to a dramatically shortened lifespan. They use electrophysiological and optical approaches to demonstrate that increased Parp1 activity (caused by Xrcc1 deficiency) triggers seizure-like activity in vivo, in slices and in isolated hippocampal neurons in vitro. Under these conditions, Parp1 inhibition and/or deletion in conditional Xrcc1 knockout mice suppresses seizures, restores electrophysiological activity and lengthens lifespan. The authors speculate as to how PARP inhibition might serve as a therapeutic approach to treatment of XRCC1-dependent neurological disease.

I found this to be an absolutely lovely paper. The model is well chosen, the problems addressed are significant and the experimental approaches are well thought out and well controlled. I was particularly impressed by the sophisticated activity measurements used - the MEAs and the modified GCaMP6 optical imaging. I literally have no changes to request (and I don't think I have said that in one my reviews in many years).

My only regret is that the authors focused all of their energies on the hippocampus and to a lesser extent on the cortex. This is logical and in the context of the large PARP1 increases in these areas, perfectly appropriate. But the human condition has a prominent cerebellar phenotype and it would have been of considerable interest to repeat some key experiments, particularly the MEA recordings, on slice preparations from this region to determine if there were lesser but still significant network changes there. Any answer would be interesting, even a negative one. For EMBO Reports, however, these additional observations are not required, and their absence does not diminish my strong enthusiasm for seeing this paper published in close to its current form.

We thank the referee for his/her support and enthusiasm. Indeed, we focused this work on the hippocampus, because of its link with seizures and our discovery that the latter are the cause of shortened lifespan. The cerebellum is of course of huge interest too, because as the referee correctly points out it is the source of the ataxia present in this mouse model (and in the associated and other related SSB repair-defective human diseases). We have presented data describing the elevated ADP-ribosylation in the cerebellum of this mouse model in our earlier paper (Hoch et al Nature, 2017), and also the presence of electrophysiological defects in cerebellar Purkinje cell (Supplementary Figure 9 in the Hoch paper). We plan to examine these in more detail using MEA in future work, when we will turn our focus once again to the ataxia phenotype.

Referee #2:

In this manuscript, Komulainen et al. show that lethal seizures and shortened lifespan due to loss of Xrcc1 in the nervous system can be rescued by deleting PARP1. The authors further show that deleting PARP1 can correct defects in presynaptic calcium signaling in Xrcc1-deficient neurons, and also decrease seizure like activity in Xrcc1-deficient hippocampal slices. The experiments are well designed and executed. Importantly, the reduction of seizure-like activity in Xrcc1Nes-Cre animals by PARP1 deletion is quite interesting and new. Addressing the comments below would further strengthen the points made in the manuscript:

1. Figure 1 shows that loss of PARP1 can cause a reduction in ADP-Ribose levels in Xrcc1 KO mice. Taken together with the reduced PARP1 staining, the authors conclude that PARP1 is hyperactive in Xrcc1 KO brains. However, the figure does not directly show that the remaining PARP1 protein in these brains is actually hyperactive. The authors say throughout the text and figures that PARP1 is aberrant/hyperactive in Xrcc1 KO brains, but this is never directly demonstrated biochemically. We have now addressed this question biochemically, as suggested by the referee, by incubating hippocampal and cerebellar tissue extracts with NAD⁺. We show that PARP1-mediated ribosylation is

elevated in *Xrcc1*-defective hippocampal and cerebellar tissue extracts, when compared to that present in the corresponding wild type extracts. We have added the biochemical experiments to Fig.1b,c (replacing the anti-PARP1 IF, which has been reported previously by us in Lee et al 2009) In addition, we note that the elevated ADP-ribose that is uncovered in cultured *Xrcc1*^{Nes-Cre} neurons by incubation for 1-hr with PARG inhibitor (Fig.5a) is the result of hyperactive PARP1, because this elevated ADP-ribose is prevented by PARP inhibitor. This is discussed in the text (bottom of Page 6).

2. Related to the point above, the quantification in Figure 1 shows that the loss of PARP1 causes a general reduction in ADP-Ribose levels unrelated to the loss of *Xrcc1* (blue vs gray bars). This could account for the lack of detectable ADP-ribose staining instead of a specific PARP1 hyperactivation in *Xrcc1* KO brains. Both points could be addressed by assessing the biochemical activity of PARP1 from these tissues. I

I apologise, I'm not sure if I understand this point. ADP-ribose staining is elevated in the *Xrcc1*^{Nes-Cre} brain, and is suppressed as expected by additional deletion of one or both alleles of *Parp1* (Fig.1a). However, we have as requested added the suggested biochemical experiments (Fig.1b,c, and see above).

3. Several correlations described by the authors need further clarification. For instance, in Figure 2, the observation is made that the loss of a single PARP1 copy is protective. However, in Figure 1, poly-ADP ribose levels are significantly elevated under these conditions. On the other hand, whereas the loss of PARP1 alone does not seem to affect mortality, its loss in the context of *Xrcc1* seems toxic. Based on these results, the correlation between PARP1 levels, PARP1 activity, and mortality seem unclear. Finally, the loss of a single copy of PARP1 is sufficient to reduce seizures even though this PARP1 would seem to be hyperactive based on the authors interpretations of Figure 1.

For the phenotypes of elevated ADP-ribose levels, dysfunctional calcium signalling, and elevated seizures (whether measured by electrophysiology or by video imaging) the relationship with *Parp1* is straightforward. In *Xrcc1*^{Nes-Cre} brain, the deletion of one *Parp1* allele partially rescues these phenotypes and the deletion of both *Parp1* alleles fully rescues them. However, we agree that the relationship between *Parp1* genotype and lifespan is complex. The confusing result is that deletion of one *Parp1* allele rescues lifespan in *Xrcc1*^{Nes-Cre} mice more than does deletion of both *Parp1* alleles. Our interpretation of this is that whilst deletion of both *Parp1* alleles ablates the elevated ADP-ribose levels, calcium dysfunction, and seizure-induced death in *Xrcc1*^{Nes-Cre} mice, the complete absence of *Parp1* imposes a new (as yet undefined) defect; resulting in mice that live longer than *Xrcc1*^{Nes-Cre} mice (because the seizure-dependent death is prevented) but younger than *Xrcc1*^{Nes-Cre} mice that retain one *Parp1* allele. We do not yet know why it is important to retain one *Parp1* allele in *Xrcc1*^{Nes-Cre} mice, but it is clearly related to the absence of *Xrcc1* because as the referee noted single *Parp1* KO mice have a normal lifespan. It will now be of great interest to identify the mechanism of this (seizure-independent) death in the double KO mice. This is now discussed on Page 5 & 9 of the revised manuscript.

4. The results in Figure 4 showing the reduction of seizures following the loss of PARP1 in *Xrcc1* KO mice is indeed interesting. Does the loss of PARP1 also affect the ability to induce seizures in a WT background? The same comment extends to the results shown in Figure 6. Does the loss of PARP1 affect calcium signaling in a WT background?

This is a great question. We have now conducted some initial analysis of our current *Parp1*^{-/-} animals, but cannot see a difference in either seizure frequency by MEA or in calcium signalling (see the Figure below). This suggests that *Parp1* activity does not contribute to these phenotypes in wild type mice, at least at the juvenile mouse age we have examined. If the Referee agrees, we would rather not include this negative data in the manuscript because it might be that the situation is different if we examine aged *Parp1*^{-/-} mice (e.g. 1-2 yr old aged mice), in which endogenous levels of DNA damage have had a chance to accumulate. That would be really exciting but is beyond the time-frame of this manuscript.

[a] Acute brain slices of the indicated genotypes were recorded on MEA. Mean cumulative activity plots in the CA3 region of hippocampus over 10 min of recording in epileptogenic buffer. *WT* (n = 13 slices from 3 mice), *Xrcc1^{Nes-Cre}* (n = 9, 3), *Parp1^{-/-}* (n=3, 1). **[b]** Mean SyGCaMP6f calcium responses of cultured hippocampal primary neurons to three rounds of 10 APs stimulation from mice of the following genotypes; *WT* (n = 1257 synapses, 5 coverslips, 2 animals), *Xrcc1^{Nes-Cre}* (n =3313,12, 4)* *Parp1^{-/-}* (n = 1251, 5, 2). *Data transposed from Fig.6 of the manuscript for comparison.

5. The results in Figure 5 further suggest the importance of directly assessing PARP1 activity in WT and *Xrcc1* KO neurons and tissues.

This we have now done, as suggested above.

Referee #3:

Komulainen et al. investigated the mechanism linking defects in DNA SSB with neurological dysfunction. The work showed hyperactivity of Parp1 in *XRCC1NES-CRE* mice results in lethal seizures and shortened lifespan. And both defects are prevented by Parp1 inhibition. Overall, the work is of interest and well written, and highlights PARP inhibition as a possible therapeutic approach in *XRCC1*-mutated neurological disease.

My comments are as follows:

1. The authors checked Parp1 protein level, as well as ADP-ribose in mice brains. It's better to show PARylation in neurons and tissues.

We employed a pan-ADP-ribose detection reagent for most of this work because of the sensitivity and reliability of the reagent. However, we have also detected the elevated ADP-ribosylation in *Xrcc1^{Nes-Cre}* mouse brain sections by IHC using anti-poly(ADP-ribose) antibodies (Figure EV1) and in *Xrcc1^{Nes-Cre}* tissue extracts by WB using poly(ADP-ribose)-specific detection reagent (Fig.1b,c). We also note that, in cultured neurons at least, we require a short incubation with PARG inhibitor to detect the elevated ADP-ribose, confirming that this signal is also poly(ADP-ribose) (Fig.5a).

2. How Parp1 hyperactivity affects Ca²⁺ signaling, one possibility is that this is a result of NAD⁺ depletion. The authors should measure NAD⁺ levels in the cells and tissues.

Indeed, we have now measured NAD⁺ levels and find that these are reduced by ~50% in *Xrcc1^{Nes-Cre}* brain. We have added these data to Fig.1d and discussed this finding as a possible explanation for the calcium signaling defect on Page 10.

3. The lifespan and seizure results are good. Did you perform other behaviour assays on the mice? Such as cognition and motor function assays, which are closely related to neurological diseases.

Since this work is focused on the hippocampal/seizure phenotype and its impact on mortality, we have not conducted other behavioural tests. However, we have tested motor function and described the ataxia in this mouse model in our previous work (Lee et al Nat. Neuroscience 2009; Hoch et al, Nature 2017).

4. Besides Parp1, it's better to measure other DNA repair related proteins in *XRCC1NES-CRE* and *Parp1^{-/-}* *XRCC1NES-CRE* mice brains.

As requested, we have now added IHC of another DNA repair protein (Atm), in Figure EV1

5. Parp1 inhibition seems to be a potential therapeutic target for XRCC1-mutated neurological disease. Then if treated XRCC1NES-CRE mice with Parp1 inhibitor, will the longevity and seizure be improved?

We have now conducted these experiments as suggested. We now show that, in addition to correcting the defect in calcium signaling in cultured neurons, Parp1 inhibitor also prevents the elevated seizure-like activity in *Xrcc1^{Nes-cre}* brain slices, as measured by MEA (Fig.4e). This is an exciting result. This was achieved by inclusion of the inhibitor in the drinking water of the mother, since the tissues were extracted and analysed prior to weaning. This is an exciting finding, because it supports the possibility that PARP inhibition might provide a therapeutic approach for the treatment of XRCC1-defective, and possibly other, neurological diseases. However, we have not yet observed lifespan rescue. Currently available inhibitors may not be suitable for this purpose, because they 'trap' PARP enzymes on unrepaired SSBs and thereby exacerbate the DNA repair defect in SSB repair-defective cells, causing increased DNA replication fork stalling and/or collapse during S phase. Whilst this is not a problem for post-mitotic neurons, it is likely to be cytotoxic in proliferating neural and other cell types. Indeed, proliferating XRCC1-defective cells are hypersensitive to current PARP inhibitors, perhaps explaining we have so far been unable to extend significantly the lifespan in *XRCC1^{Nes-Cre}* mice. We have discussed this in the text (Page 10/11).

6. PARP1 and NAD+ axis in ageing-related disease and neurodegenerative diseases should be discussed, there are several relevant papers and reviews in the literature.

Thanks you, we have now discussed this in the text (Page 10).

Dear Keith,

Thank you for the submission of your revised manuscript. We have now received the enclosed reports from the referees that were asked to assess it, and I am happy to say that both support its publication now. Only a few more minor editorial changes will be required before we can proceed with the official acceptance of your manuscript.

- Please add up to 5 keywords to the manuscript.
- Please correct the subheading to "Conflict of interest".
- Please add all author contributions.
- I attach to this email a related manuscript file with comments by our data editors. Please address all comments in the final manuscript (e.g. the number of replicates is not clearly defined in Fig 3C & Fig 6G).
- The manuscript sections need re-ordering, please move the figure legends to after the references.

EMBO press papers are accompanied online by A) a short (1-2 sentences) summary of the findings and their significance, B) 2-3 bullet points highlighting key results and C) a synopsis image that is exactly 550 pixels wide and 200-600 pixels high (the height is variable). You can either show a model or key data in the synopsis image. Please note that text needs to be readable at the final size. Please send us this information along with the revised manuscript.

Best wishes,
Esther

Referee #2:

The authors have addressed all my concerns satisfactorily. I think the manuscript is suitable for publication in EMBO reports without additional revision.

Referee #3:

The authors have now adequately revised and the paper is suitable for publication

The authors have addressed all minor editorial requests.

Prof. Keith Caldecott
University of Sussex
Genome Damage and Stability Centre
Science Park Road
Falmer
Brighton, Sussex BN1 9RQ
United Kingdom

Dear Prof. Caldecott,

I am very pleased to accept your manuscript for publication in the next available issue of EMBO reports. Thank you for your contribution to our journal.

At the end of this email I include important information about how to proceed. Please ensure that you take the time to read the information and complete and return the necessary forms to allow us to publish your manuscript as quickly as possible.

As part of the EMBO publication's Transparent Editorial Process, EMBO reports publishes online a Review Process File to accompany accepted manuscripts. As you are aware, this File will be published in conjunction with your paper and will include the referee reports, your point-by-point response and all pertinent correspondence relating to the manuscript.

If you do NOT want this File to be published, please inform the editorial office within 2 days, if you have not done so already, otherwise the File will be published by default [contact: emboreports@embo.org]. If you do opt out, the Review Process File link will point to the following statement: "No Review Process File is available with this article, as the authors have chosen not to make the review process public in this case."

Should you be planning a Press Release on your article, please get in contact with emboreports@wiley.com as early as possible, in order to coordinate publication and release dates.

Thank you again for your contribution to EMBO reports and congratulations on a successful publication. Please consider us again in the future for your most exciting work.

THINGS TO DO NOW:

You will receive proofs by e-mail approximately 2-3 weeks after all relevant files have been sent to our Production Office; you should return your corrections within 2 days of receiving the proofs.

Please inform us if there is likely to be any difficulty in reaching you at the above address at that time. Failure to meet our deadlines may result in a delay of publication, or publication without your corrections.

All further communications concerning your paper should quote reference number EMBOR-2020-51851V3 and be addressed to emboreports@wiley.com.

Should you be planning a Press Release on your article, please get in contact with emboreports@wiley.com as early as possible, in order to coordinate publication and release dates.

Corresponding Author Name: Keith Caldecott

Manuscript Number: EMBOR-2020-51851-T